# Description of *Pegethrix niliensis* sp. nov., a Novel Cyanobacterium from the Nile River Basin, Egypt: A Polyphasic Analysis and Comparative Study of Related Genera in the Oculatellales Order

**DOI:** 10.3390/toxins16100451

**Published:** 2024-10-21

**Authors:** Guilherme Scotta Hentschke, Zakaria Mohamed, Alexandre Campos, Vitor M. Vasconcelos

**Affiliations:** 1Interdisciplinary Centre of Marine and Environmental Research, University of Porto, Terminal de Cruzeiros de Leixões, Av. General Norton de Matos s/n, 4450-208 Matosinhos, Portugal; amoclclix@gmail.com (A.C.); vmvascon@fc.up.pt (V.M.V.); 2Department of Botany and Microbiology, Faculty of Science, Sohag University, Sohag 82524, Egypt; zakaria.attia@science.sohag.edu.eg; 3Faculty of Sciences, University of Porto, Rua do Campo Alegre s/n, 4069-007 Porto, Portugal

**Keywords:** biodiversity, new species, 16S rRNA gene, microcystin, BMAA

## Abstract

In this paper, we examine the filamentous cyanobacterial strain NILCB16 and describe it as a new species within the genus *Pegethrix*. The original population was sampled from a mat growing in an irrigation canal in the Nile River, Egypt. Initially classified under *Plectonema* or *Planktolyngbya,* the strain is a potential producer of the toxins microcystin and β-N-Methylamino-L-Alanine (BMAA). Additionally, we reviewed the taxonomic relationships between the Oculatellales genera. To describe the new species, we conducted a polyphasic study, encompassing 16S rRNA gene phylogenetic analyses performed using both Maximum Likelihood and Bayesian methods, sequence identity (p-distance) analysis, 16S-23S ITS secondary structures, and morphological and habitat comparisons. The phylogenetic analysis revealed that strain NILCB16 clustered within the *Pegethrix* clade with strong phylogenetic support, but in a distinct position from other species in the genus. The strain shared a maximum 16S rRNA gene identity of 97.3% with *P. qiandaoensis* and 96.1% with the type species, *P. bostrychoides*. Morphologically, NILCB16 can be differentiated from other species in the genus by its lack of false branching. Our phylogenetic analyses also show that *Pegethrix*, *Cartusia*, *Elainella*, and *Maricoleus* are clustered with strong phylogenetic support. They exhibit high 16S rRNA gene identity and are morphologically indistinguishable, suggesting they could potentially be merged into a single genus in the future.

## 1. Introduction

Cyanobacteria are photosynthetic and prokaryotic microorganisms that initially evolved about 3500 million years ago [1]. They are found in a wide range of habitats, including in terrestrial, marine, and freshwater environments, as well as in harsh environments like hot springs [2,3,4]. Cyanobacteria play a vital role in the global ecosystem by producing oxygen and fixing carbon and nitrogen [5]. Cyanobacteria present a complex and challenging taxonomy, with numerous cryptic taxa being frequently misidentified [6].

Classification of cyanobacteria based on their morphological characteristics has led to taxonomic confusion, because of their simple structure and the similar characteristics among different taxa [7]. However, the emergence of the polyphasic taxonomic approach, which combines morphological, eco-physiological, biochemical, and molecular traits, resulted in substantial revisions of cyanobacterial classification and the establishment of new orders, families, genera, and species [8,9,10].

An example of this can be observed in the order Oculatellales, where strains previously assigned to *Leptolyngbya* by morphological analysis have been reclassified using a polyphasic approach into numerous clades and new genera such as *Drouetiella* [11], *Thermoleptolyngbya* [3], *Timaviella* [11], *Trichotorquatus* [12], and *Pegethrix* [10].

Among the recently described genera of this order, *Pegethrix* has seven characterized species including *Pegethrix botrychoides*, *P. olivacea*, *P. convoluta*, *P. indistincta* [11], *P. atlantica* [13], *P. sichuanica* [14], and *P. qiandaoensis* [10]. Nevertheless, the majority of *Pegethrix* species are geographically confined to the USA [11,13], with two species reported from China [10,14] and one from the Azores Islands, Portugal [13]. Nevertheless, strains isolated from various regions of the world that had morphological traits like *Pegethrix*, but were mistakenly assigned to *Lyngbya*/*Phormidium*/*Plectonema* and *Leptolyngbya* [9,11], should be re-evaluated using the polyphasic approach in order to ascertain their exact phylogenetic position.

In Egypt, the irrigation canals split from the Nile River, especially those located in a subtropical region, have been influenced by high temperature, humidity, intense sunlight exposure, and increasing anthropogenic activities [15]. These conditions promoted the proliferation of microalgae and resulted in frequent harmful cyanobacterial blooms in these water bodies during warm months every year [16]. The cyanobacterial communities in some irrigation canals in Upper Egypt are dominated by a *Leptolyngbya*-related morphotype, and based on morphological characteristics, this cyanobacterium was previously reported under *Plectonema* [17] or *Planktolyngbya* [18]. Interestingly, ELISA and HPLC analyses linked this cyanobacterium to the production of microcystin –YR and –LR [17]. Furthermore, the toxin β-N-Methylamino-L-Alanine (BMAA) was also reported in mats where this cyanobacterium was collected [17,18].

Therefore, the main aim of this study is to apply a polyphasic approach to clarify the phylogenetic position of a *Leptolyngbya*-related morphotype, which was isolated (strain NILCB16) from a Nile River irrigation canal. The present study determined the position of this cyanobacterium within the genus *Pegethrix*, as the new species *Pegethrix niliensis* sp. nov. Additionally, this paper reviews the close phylogenetic relationship among *Pegethrix, Cartusia*, *Elainella*, *Maricoleus,* and *Drouetiella*.

## 2. Results


Description of the new taxonOrder OculatellalesFamily Oculatellaceae*Pegethrix niliensis* G. S. Hentschke sp. nov.
Figure 1



In liquid medium, growing cespitose from the bottom of the flask, forming erect fascicles, or with filaments attached to the flask walls. Filaments long, straight, flexuous or wavy, entangled or forming fascicles. Sheaths firm, colorless, tightly embracing the trichome (up to 1 μm) or widened (up to 3 μm), containing only one trichome. Trichomes facultatively constricted, isopolar, cylindrical. Apical cells rounded. Cells isodiametric, longer or shorter than wide. Cell content homogenous or granulose. Cell measurements: 1.8–3.5 μm long and 2.3–2.7 μm wide.

Etymology: *niliensis* is derived from the Latin word “Nilus”, which refers to the Nile River. The suffix “-ensis” indicates origin or place.

Holotype: A metabolically inactive (lyophilized) biomass of the strain NILCB16 was deposited at the microorganism’s herbarium of the Faculty of Science, Sohag University, under the number SOU00197.

Type strain: NILCB16 (PQ314731).

Habitat: Mat at the margin of a freshwater irrigation canal from Nile River, Egypt.

**Figure 1 toxins-16-00451-f001:**
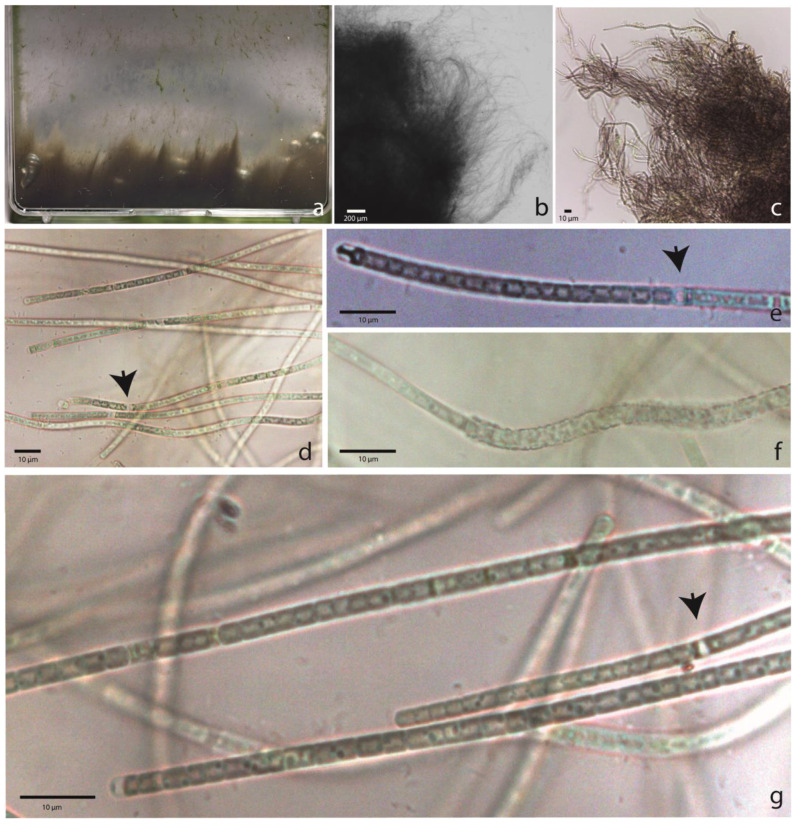
*Pegethrix niliensis* sp. nov. (**a**) Thalli growing in liquid medium. (**b**,**c**) Filaments with entangled arrangement and forming fascicles. (**d**) Details of trichomes. (**e**) Constricted filament with necridium. (**f**) Filament with widened sheath. (**g**) Details of trichomes. Arrows indicate necridia.

The first-round phylogeny (Figure 2) demonstrated that the order Oculatellales was monophyletic (ML = 100), encompassing the genera *Oculatella*, *Tildeniella*, *Calenema*, *Shahulinema*, *Thermoleptolyngbya*, *Drouetiella*, *Pegethrix*, *Cartusia*, *Maricoleus*, *Elainella*, *Albertania*, *Egbenema*, *Trichotorquatus*, *Komarkovaea*, *Timaviella,* and *Shackletoniella*. Among these, the genera *Pegethrix*, *Elainella*, *Cartusia*, and *Maricoleus* formed a highly supported cluster (ML = 100), indicating a close phylogenetic relationship among them. *Drouetiella* was positioned as a sister group to this cluster (ML = 0.8).

This phylogeny was also able to separate the *Pegethrix* species in distinct clades. Our isolate *P. niliensis* sp. nov. (labeled as SH) was placed at the base of the *Pegethrix* clade, along with “*Leptolyngbya*” WR9, with robust phylogenetic support (ML = 90), suggesting a close relationship with this genus. However, *P. niliensis* sp. nov. was phylogenetically distinct from the clades of the already established species within the genus, suggesting that it is a new taxon worthy to be described. The genus *Elainella* was found to be polyphyletic, as *E. chongqingensis* CCNU0012 was phylogenetically separated from the type species strain *E. saxicola* UPOC E1. Between these two *Elainella* species, the *Maricoleus* and *Cartusia* clades were placed.

The second-round ML and BI phylogenies confirmed the polyphyletic status of *Elainella* (Figure 3). As in the first-round analysis, the second-round analysis showed the genera *Pegethrix*, *Elainella*, *Cartusia*, *Maricoleus,* and *Drouetiella* forming a strongly supported cluster (ML = 91, BI = 1). The *Pegethrix* clade was robustly defined with strong statistical support (ML = 96, BI = 1). In these analyses, as in the first-round tree, *P. niliensis* sp. nov. was phylogenetically separated from the other species within the genus, suggesting that it represents a new taxon worthy of description.

The 16S rRNA gene identity analysis confirmed a close relationship among the genera *Pegethrix*, *Elainella*, *Cartusia*, *Maricoleus*, and *Drouetiella*. Comparisons between their respective clades revealed that their 16S rRNA identity values were frequently above 95% (Appendix A). When comparing only the reference/type strains, *Pegethrix bostrychoides* GSE-PSE-MK47-15B shared 16S rRNA gene identity values of 95.8% with *Elainella saxicola* UPOC E1, 95.7% with *Cartusia fontana* Kovacik 1999/1-LC, 94.9% with *Maricoleus vaginatus* WZU 0102, and 94% with *Drouetiella lurida* LUKESOVA1986/6.

*P. niliensis* sp. nov. also exhibited high 16S rRNA gene identity with all these genera, with values ranging from 94% to 97.6% with *Pegethrix*, 96.8% to 97.6% with *Cartusia*, 96.3% to 97.6% with *Elainella*, 95.1% to 95.9% with *Drouetiella* and 96.8% with *Maricoleus*. 

The 16S-23S ITS secondary structure analysis aligned with the previous analyses, indicating close relationship among all these genera (Figure 4, Figure 5, Figure 6 and Figure 7). All of them presented the sequence 5′-CAUCCCA-3′ in the basal lateral bulge of the D1-D1′ helix, with the only exception of *P. altantica*, which showed a single substitution resulting in the sequence 5′-CAUCUCA-3′ (Figure 4). No diagnostic characters in the secondary structures were found to separate these genera. In contrary, the sequence 5′-CAUCCCA-3′ at the lateral bulge can be considered a constant character for them. As for all the compared genera, the D1-D1′ helix of *P. niliensis* sp. nov. also featured the sequence 5′-CAUCCCA-3′ at the lateral bulge. However, the remaining regions of this helix, as well as the V2, Box B, and V3 helices, differed from the others in sequence, length, and structure.

The V2, Box B, and V3 helices showed differences in sequence, length, and structure across the compared genera (Figure 5, Figure 6 and Figure 7). This variability was also evident among the *Pegethrix* species, which displayed significant differences in sequence, length, and structure across all ITS helices.

Morphologically, *Pegethrix*, *Cartusia*, *Elainella*, and *Drouetiella* are quite similar and cannot be distinguished. They all exhibit entangled or fascicled filaments. The trichomes contain necridia, may be facultatively constricted, and exhibit apical cells rounded, not tapered. Their sheaths are firm and colorless. Their cells are isodiametric, longer or shorter than wide, with granulose content. All these genera inhabit humid (freshwater) terrestrial environments. *Maricoleus* is also morphologically similar but is a marine genus (Table 1). *P. niliensis* sp. nov. is from freshwater and fits into the description of any of these genera, presenting the same morphological characters. At the species level, we compared *P. niliensis* sp. nov. with all *Pegethrix* species, as shown in Table 2. It was evident that, morphologically, all *Pegethrix* species are similar, but present some differences, as detailed in the Discussion section.

All of our results considering 16S rRNA gene phylogenetic analysis, identity, 16S-23S ITS secondary structures, and morphological and habitat analyses indicate that *Pegethrix*, *Cartusia*, *Maricoleus*, *Elainella,* and *Douetiella* are very closely related genera. However, our isolate was phylogenetically clustered to *Pegethrix* and fit into the circumscription of this genus, and because of that, we herein describe the new species *Pegethrix niliensis* sp. nov.

## 3. Discussion

Although the primary objective of this paper is to describe a new species of *Pegethrix*, our analysis yielded two other significant findings: (1) *Pegethrix*, *Cartusia*, *Elainella,* and *Maricoleus* are clustered together, present high 16S rRNA gene identity values, and are morphologically indistinguishable from each other and (2) *Elainella* is currently polyphyletic.

To address these issues, two potential solutions are possible: (1) split *Elainella* into two genera or (2) merge the above-cited genera into a single monophyletic genus. We advocate in favor that in the future *Pegethrix*, *Cartusia*, *Maricoleus*, *and Elainella* should be merged into a single genus for the reasons discussed below. The case of *Drouetiella* is more complicated as it shows lower 16S rRNA gene identity values when compared to the other genera (Appendix A) and probably warrants classification as a separate genus.

We conducted a polyphasic analysis and found that *Pegethrix*, *Cartusia*, *Maricoleus*, and *Elainella* form a monophyletic cluster (Figure 2 and Figure 3). Within this cluster, the 16S rRNA gene identity values were high, with all genera sharing more than 95% identity. According to the phylogenetic results, the morphological comparisons, the monophyletic concept of cyanobacterial genera [19], and the 16S rRNA gene threshold of Yarza et al. [20], which suggests that strains with more than 94.5% 16S rRNA gene identity should belong to the same genus, there is no justification for the separation of these genera. Moreover, their habitats are similarly characterized by humid rocky walls or freshwater environments, except for *Maricoleus*, which is marine. Another piece of evidence is that the 16S-23S ITS secondary structures of the D1-D1′ helix of all the genera featured a conserved region, 5′-CAUCCCA-3′, in the lateral bulge. Furthermore, no diagnostic region in these structures is present to distinguish these genera (e.g., there is no unique region among the *Pegethrix* helices which can distinguish it from the other genera). These results align with the original publications of these genera and their species [11,21,22], which also demonstrated that this group is monophyletic, with high 16S rRNA gene identity and very similar morphological and ecological traits. However, they were treated as different taxa.

In their original description, *Pegethrix*, *Cartusia*, and *Drouetiella* were distinguished among them and from *Elainella* solely based on 16S-23S ITS secondary structures and the length of certain 16S-23S ITS domains [11]. However, we believe this criterion is not robust, as our analysis revealed that these genera share significant similarities in their D1-D1′ helices, particularly in the conserved region of the lateral bulge. Furthermore, if the 16S-23S ITS regions are used as a diagnostic criterium to separate these genera, then logically, all *Pegethrix* species would also need to be classified as different genera, given the significant variation among their helices (Figure 4, Figure 5, Figure 6 and Figure 7). Additionally, the 16S rRNA gene identities among the *Pegethrix* species are sometimes lower than those observed between strains from different genera. For example, the *Pegethrix* strains *Pegethrix* FACHB3566 and *P. olivacea* GSEPSEMK4615A share 96.37% of 16S rRNA gene identity, while strains from different genera, such as *Cartusia fontana* KOVACIK1999/1 and *P. indistincta* GSETBCMK07GA, share 96.47% of 16S rRNA gene identity. In our analyses, we could not include *Cartusia* and *P. qiandoaensis* 16S-23S ITS secondary structures, because their sequences are not available in public databases.

We also highlight that interpreting 16S-23S ITS secondary structures can be highly subjective, with some researchers identifying differences, while others find similarities among them, as seen in this case. The problem of interpretation of secondary structures is discussed by Oliveira et al. [23], who demonstrated that genera from different orders can also exhibit similar 16S-23S ITS secondary structures. Additionally, some studies [24,25,26] showed that different operons within the same strain can have varying lengths in their conserved domains, complicating comparisons even more. According to Jusko et al. [27], “analyses of the 16S–23S ITS region are highly subject to the effects of multiple ribosomal operons in a genome and evidence is mounting that widespread issues may have arisen due to the lack of consideration of this phenomenon” [25]. For the reasons presented in our article, we believe that *Pegethrix*, *Cartusia*, *Elainella*, and *Maricoleus* should be merged in the future. In this case, the name adopted should be *Elainella*, which is the oldest genus [28]. *Drouetiella*, however, exhibits lower 16S rRNA gene identity values with these genera and possibly warrants classification as a separate genus.

Regarding the species *P. niliensis* sp. nov., it is phylogenetically clustered to *Pegethrix* with strong statistical support, and because of that, we assigned it to this genus. After this assignment at the genus level, we also found that *P. niliensis* sp. nov. was phylogenetically distinct from previously described species of *Pegethrix*. The 16S rRNA gene identity matrix (Appendix A) indicated that the identity values shared between *P. niliensis* sp. nov. and other *Pegethrix* species were always below 98.7%, the threshold for species delimitation as defined by Yarza et al. [20]. According to these authors, strains with 16S rRNA gene identity values below this threshold are considered to represent different species. The highest 16S rRNA gene identity value was found with *Pegethrix* FACHB3566, which was only 97.6% identical to *P. niliensis* sp. nov.

Morphologically, *P. niliensis* sp. nov. differs from *P. bostrychoides*, *P. olivacea*, *P. qiandaoensis,* and *P. atlantica* in that it lacks false branching. *P. indistincta* is unique among these species in that it can present more than one trichome within a single sheath. While *P. niliensis* sp. nov. is morphologically more similar to *P. sichuanica*, it can be distinguished from this species through 16S rRNA gene phylogenetic and identity analyses. The occurrence of *Pegethrix niliensis* sp. nov. as a constituent of cyanobacterial blooms/mats in Egyptian irrigation canals fits ecologically with *P. atlantica* and *P. qiandaoensis* inhabiting freshwater lakes in the Azores Islands and China, respectively. However, the freshwater habitat of this new species distinguishes it from the remaining five described species, which are primarily found in hot springs, rocks, and soil crusts (Table 2).

Another important remark of our work is that if hypothetically *Pegethrix*, *Cartusia*, *Elainella*, *Maricoleus,* and *Drouetiella* were merged, the polyphyletic status of *Elainella* would also be addressed. Our three phylogenetic analyses reveal that *E. chongqingensis* formed a distinct clade separate from the type species *E. saxicola*. In fact, *E. chongqingensis* was more closely related to *Maricoleus* and *Cartusia* than to *E. saxicola*. Although among these taxa, *Maricoleus* was the most recently described, the article that described this genus [22] did not show a phylogeny that encompassed *E. chongqingensis*, preventing a direct comparison with our findings. However, this phylogeny also presented *Pegethrix*, *Elainella*, *Maricoleus*, *Cartusia,* and *Drouetiella* as a monophyletic cluster with 16S rRNA gene identity values higher than 95%, in agreement with our results. According to that, we believe that the polyphyletic nature of *Elainella* further supports the need to review and merge these genera into a single, cohesive genus.

In conclusion, our paper represents a contribution to biodiversity research. It introduces a new species from the Nile River (Egypt), a region that is largely unexplored in terms of cyanobacterial diversity. This type strain was isolated from an irrigation canal, where toxic *Leptolyngbya*-like populations commonly form blooms and produce microcystin and BMAA [17,18]. Thus, this identification is of utmost importance for understanding bloom development, toxicity risk assessment in the aquatic environment, and water management in the region. Additionally, our study provides insights into the taxonomy of the Oculatellales, highlighting its complexity and the close relation among its genera. Notwithstanding, considering that the studied strain was sampled from an environment with common toxic blooms, further studies are needed, namely on the toxic potential of the species (toxin profiling). This information will enable an assessment of the risks of its occurrence in the environment and any eventual implications to human health.

## 4. Materials and Methods

### 4.1. Sampling, Isolation of Strains, and Morphological Analysis

The strain NILCB16 was isolated from a benthic mat collected from an irrigation canal in Sohag Province, Egypt (26°62′ N and 31°64′ E), in September 2021. This irrigation canal is a permanent branch of the Nile River and serves about 155,000 acres. This water source was classified as oligotrophic–eutrophic water body dominated by cyanobacteria (1.1 × 10^5^ cells L^−1^), with physico-chemical parameters characterized by high water temperature (29 °C), slightly alkaline pH (7.8), and high nutrient concentrations (NO_3_: 1.8 mg L^−1^; PO_4_: 0.07 mg L^−1^) during summer months [18]. The isolation was performed by inoculating a single filament from the natural population into a culture flask with Z8 liquid medium [29]. The strain is currently maintained at the Algal Culture Collection at Sohag University, in Z8 liquid medium under the following conditions: 25 °C, 16꞉8 h in a light–dark cycle, with a light intensity of 40 μmol m^−2^ s^−1^) under fluorescent light.

For morphological analysis, the strain was microphotographed and analyzed by the LEICA LAS version 4.12.0 image analysis software (Leica Microsystems Limited and CMS GmbH, Heerbrugg, Switzerland). The measurements were performed for each characteristic of the strains (20 to 30 measurements) and were carried out at various positions of the slide preparation.

### 4.2. DNA Extraction, PCR Amplification, and Sequencing

The cyanobacterial filaments were harvested from the cultures, and the total genomic DNA (gDNA) of the strains were extracted using the PureLink Genomic DNA kit (Invitrogen, Waltham, MA, USA), following the manufacturer’s instructions provided for Gram-negative bacteria. For the amplification (PCR) of the 16S rRNA gene, the primers 27SF [30] and 23SR [31] were used. The PCR reaction was performed in a Veriti Thermal Cycler (Veriti 9902, Applied Biosystems, Thermo Fisher Scientific, Singapore). The final reaction volume was 20 μL, consisting of 5.9 μL of molecular biology-grade water, 4 μL of Green GoTaq Flexi Buffer, 2 μL of MgCl_2_, 2 μL of each forward and reverse primer, 1.5 μL of deoxynucleotide triphosphate (dNTPs), 0.5 μL of bovine serum albumin (BSA), 0.1 μL of GoTaq Flexi DNA Polymerase (Promega, Madison, WI, USA), and 2 μL of genomic DNA [32]. The 16S rRNA gene sequence was obtained upon PCR amplification in the following conditions: initial denaturation at 94 °C for 5 min, followed by 10 cycles of denaturation at 94 °C for 45 s, annealing at 57 °C for 45 s, and extension at 72 °C for 2 min. This was followed by an additional 25 cycles of denaturation at 92 °C for 45 s, annealing at 54 °C for 45 s, and extension at 72 °C for 2 min, with a final extension step at 72 °C for 7 min. The PCR product was separated using 1% (*w*/*v*) agarose gel stained with SYBR Safe DNA gel stain (Invitrogen by Thermo Fisher Scientific, Waltham, MA, USA), and DNA fragment of the expected size were excised from the gel and purified using the NZYGelpure kit (Nzytech, Lisbon, Portugal), following the manufacturer’s instructions. Finally, the purified fragments were sent for sequencing (separately) with primers 359F and 781R [33], 1494R and 27SF [30], 23SR [31,34], and 1114F. The sequencing was performed by sanger dideoxy sequencing at GATC Biotech (Ebersberg, Germany), and the nucleotide sequences obtained were manually inspected for quality and assembled using the Geneious Prime 2023.2.1 software (Biomatters Ltd., Auckland, New Zealand). Before phylogenetic analysis, the sequence was checked for possible chimera formation using the DECIPHER software 2.27.2 [35]. To assess the presence and quality of the DNA obtained from extraction and PCR, we performed electrophoresis on a 1% (*w*/*v*) and 1.5% (*w*/*v*) agarose gel stained with SYBR Safe DNA gel stain (Invitrogen by Thermo Fisher Scientific, Waltham, MA, USA), respectively. The confirmation of high-molecular-weight DNA was based on the presence of clear bands observed in the gel. The sequence was deposited in GenBank (National Center for Biotechnology Information, NCBI) under the code PQ314731.

### 4.3. Phylogenetic Analysis

The phylogenetic analyses were conducted in two rounds. In the first round, we aimed to position our strains among the cyanobacterial genera. For that, we aligned the 16S rRNA gene sequences of our isolate with sequences of cyanobacterial reference strains and additional sequences retrieved from GenBank (NCBI) by BLAST. The final alignment comprised 419 sequences and 813 nucleotide informative sites. Phylogenetic reconstruction was performed using the FastTree method [36], with the bootstrap value set to the default of 1000 as per the manual. The command used to run the phylogeny was “FastTree -gtr -nt alignment_file> tree_file”. The resulting tree was edited using iTOL [37].

In the second round of analysis, to confirm that the strain NILCB16 represents a new species, we selected the cyanobacterial sequences most closely related to them. This included genera from the order Oculatellales and additional genera from related orders such as Nodosilineales and Leptolyngbyales. This selection resulted in a total of 112 sequences and 894 nucleotide positions analyzed. Then, the phylogenetic trees were built using Maximum Likelihood (ML) and Bayesian Inference (BI) analyses. GTR + G + I evolutionary model was selected by MEGA11: Molecular Evolutionary Genetics Analysis version 11 [38]. The robustness of ML tree was estimated by bootstrap percentages, using 1000 replications using IQ-Tree online version v1.6.12 [39]. The Bayesian tree was constructed in two independent runs, with four chains each, for 5 × 106 generations. Burnin fraction was set to 0.25 and sample frequency was 1000, using MrBayes [40] in Cipres Gateway [41]. The processing and visualization of these trees were conducted using FigTree v1.4.4 (http://tree.bio.ed.ac.uk/software/figtree/, accessed on 1 August 2024).

For all analyses, the sequences were aligned using MAFFT [42] and the outgroup used was *Gloeobacter violaceus* PCC 8105 (AF132791). A similarity (p-distance) matrix was generated using MEGA11, and the 16S-23S ITS secondary structures of D1-D1′, V2, Box B, and V3 helices were folded using MFold [43], according to [44].

## Figures and Tables

**Figure 2 toxins-16-00451-f002:**
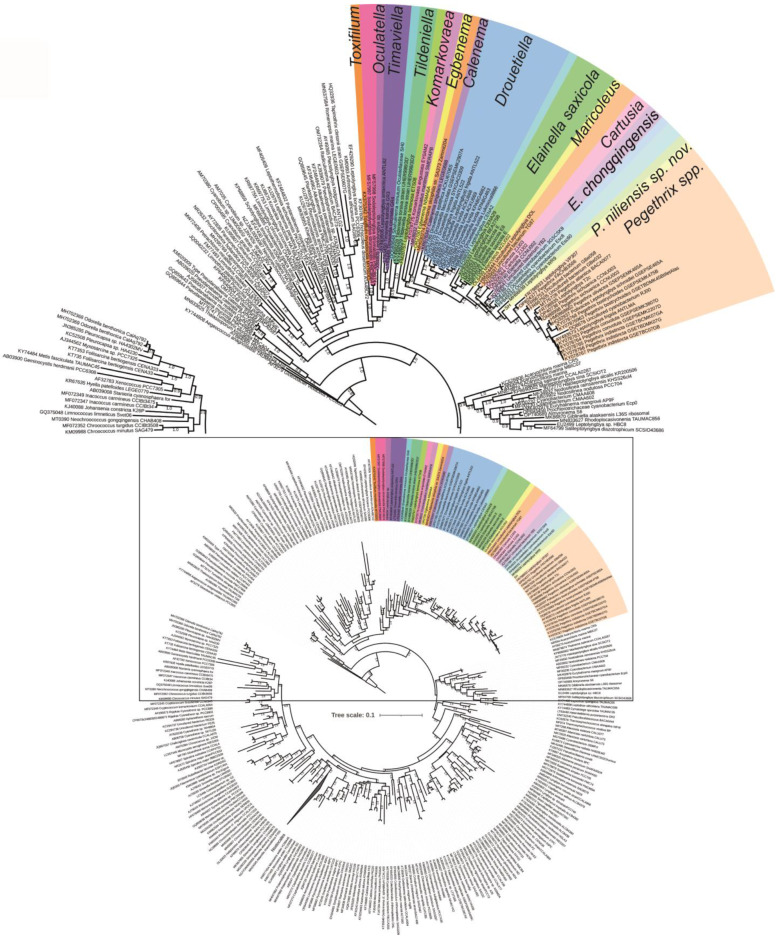
First-round phylogeny. FastTree phylogeny with cyanobacterial reference strains. The genera of Oculatellales are colored. The upper image is a section of the circular tree.

**Figure 3 toxins-16-00451-f003:**
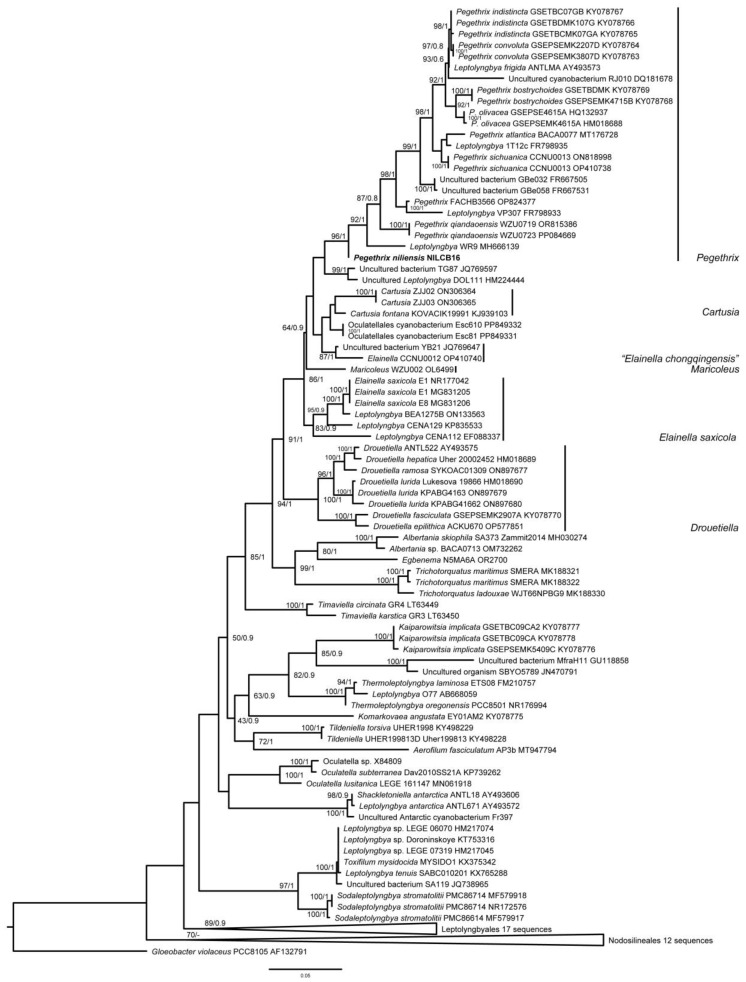
Second-round phylogeny based on Maximum Likelihood. The bootstrap values and the Bayesian posterior probabilities are indicated at the nodes.

**Figure 4 toxins-16-00451-f004:**
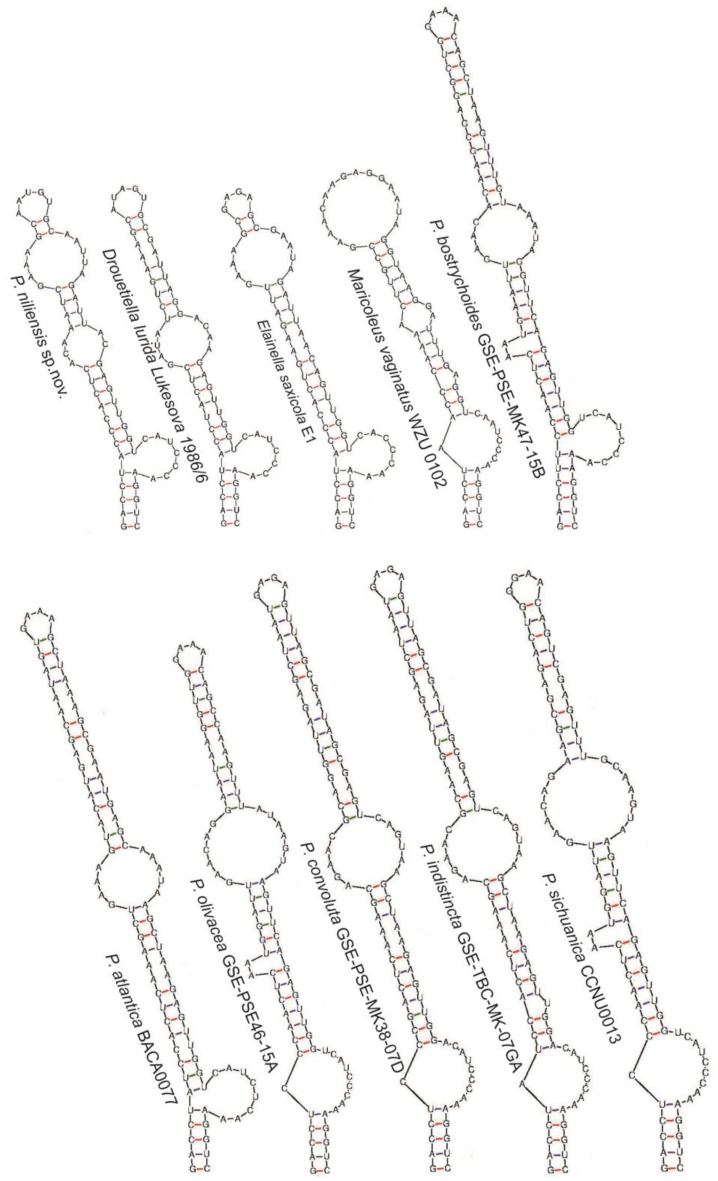
D1-D1′ helix of the 16S-23S ITS region of *P. niliensis* sp. nov. and its closest related taxa.

**Figure 5 toxins-16-00451-f005:**
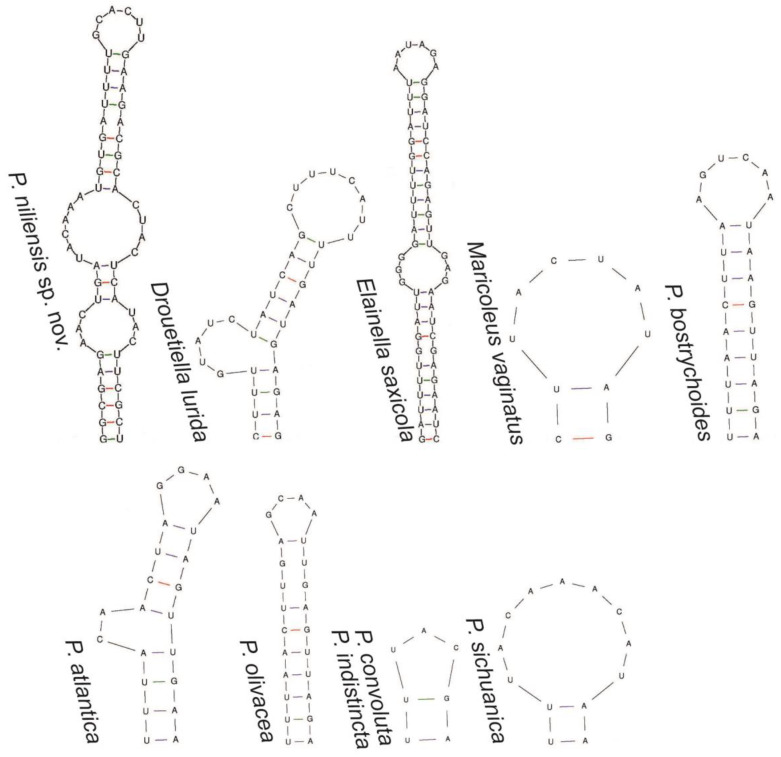
V2 helix of the 16S-23S ITS region of *P. niliensis* sp. nov and its closest related taxa.

**Figure 6 toxins-16-00451-f006:**
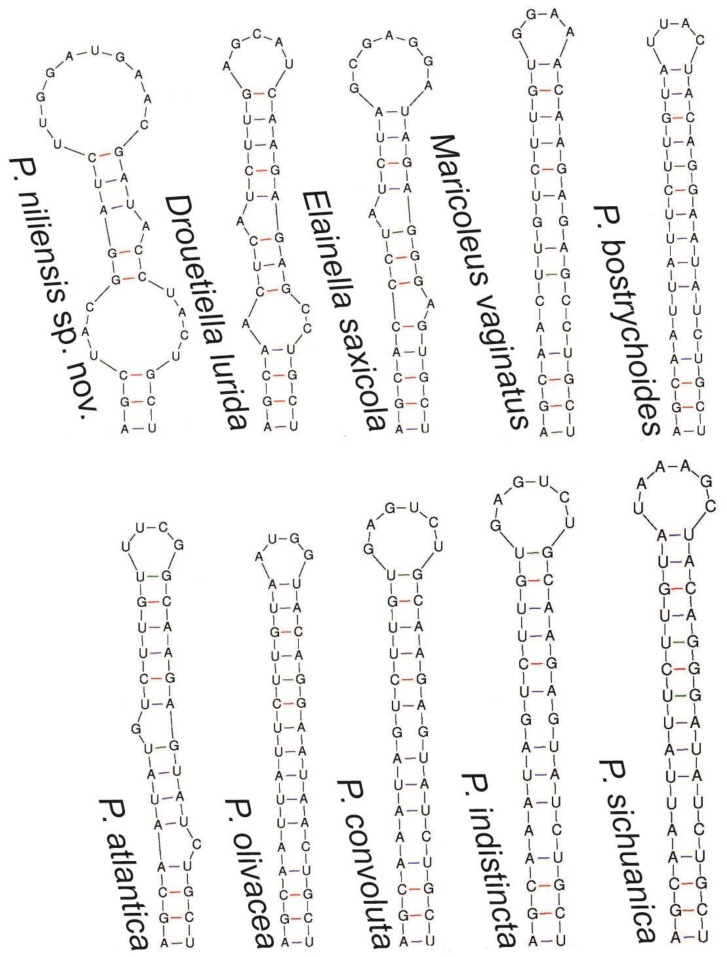
Box B helix of the 16S-23S ITS region of *P. niliensis* sp. nov. and its closest related taxa.

**Figure 7 toxins-16-00451-f007:**
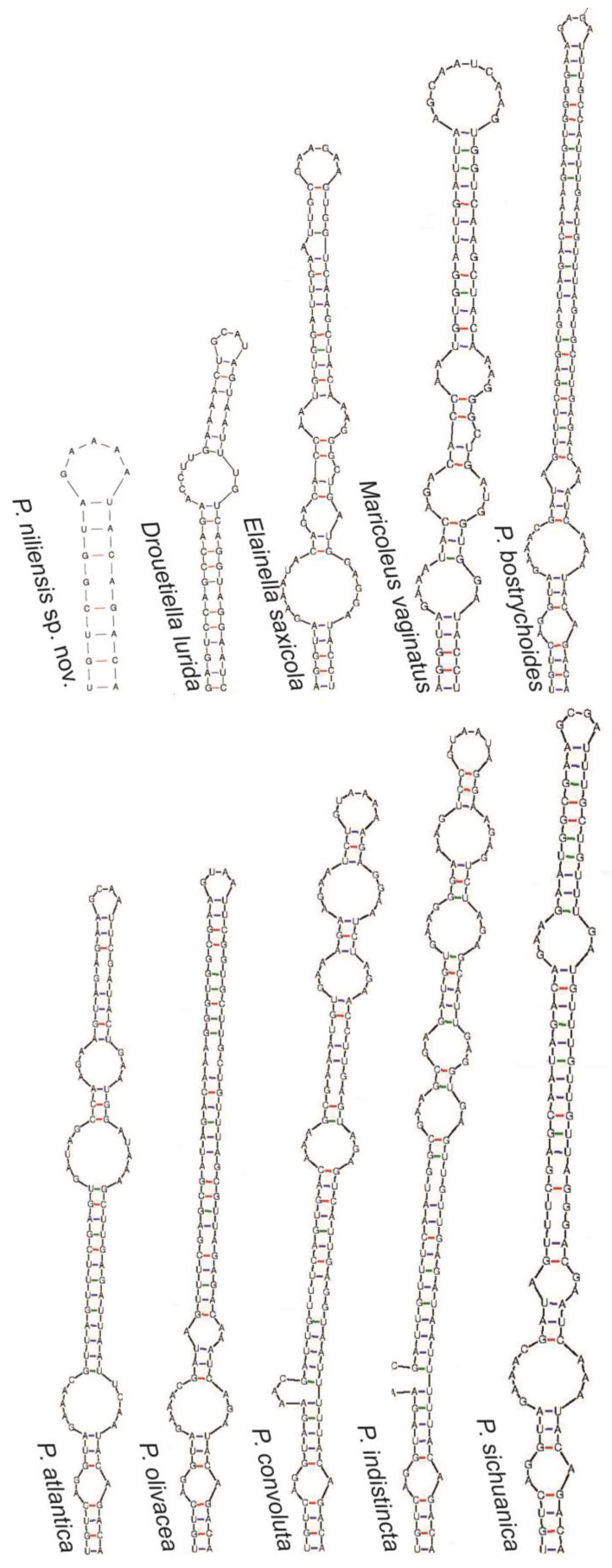
V3 helix of the 16S-23S ITS region of *P. niliensis* sp. nov. and its closest related taxa.

**Table 1 toxins-16-00451-t001:** Morphological and habitat comparisons between *P. niliensis* sp. nov. and other phylogenetically and morphologically related genera.

	*P. niliensis* sp. nov.	*Pegethrix*	*Cartusia*	*Maricoleus*	*Elainella*	*Drouetiella*
Trichome/filaments mode of growth	Erect fascicles from the bottom of the culture flasks (liquid medium)	Radially spreading	Not described	Benthic masses	Fascicles or tufts	Not described
Filaments	Long, wavy, intensely entangled or forming fascicles	Entangled or in fascicles, with nodule formation. Straight, flexuous, sometimes coiled, with nodules.	Sometimes in fascicles. Straight or flexuous, sometimes with more than one trichome in a sheath	Entangled, rarely with more than one trichome within a sheath	Forming fascicles	Solitary or in fascicles. Straight, flexuous, or coiled
Trichomes	Not tapered. Apical cells rounded	Not tapered. Apical cells rounded	Not tapered. Apical cells rounded	Not tapered	Not tapered.	Not tapered. Apical cells
False branching	Not present	Rare, single or double	Not present	Facultative	Yes, single and double	Rare, single
Constrictions at cell walls	Facultative	Facultative	Facultative	Facultative	Facultative	Facultative
Sheaths	Firm, colorless, attached to the trichome	Firm to soft and thin or widened, colorless	Firm, thin or widened, colorless	Facultative, layered, colorless, thin or widened	Firm, thin, colorless	Firm, thin, colorless
Motility	No	Yes	Not described	Not described	No	Not described
Cell shape	Isodiametric, longer or shorter than wide	Isodiametric, longer or shorter than wide	Isodiametric or shorter than wide	Isodiametric	Isodiametric or longer than wide	Isodiametric, longer or shorter than wide
Cell content	Homogenous or with granule	Homogenous or with granule	Homogenous or with granule	Granular	Often with granules	Homogenous or with granule
Necridia	Yes	Yes	Yes	Not described	Yes	Facultative
Cell dimensions (μm)	2.1–3.1 long × 1.9–2.8 wide	1–3 long × 1.3–3.3 wide	1.3–2 long × 1.8–3.5 wide	1.3–5.4 long × 1.4–4.2	1.7–2.6 long × 1.3–3.8 wide	2.1–5.4 long × 1.7–2.1 wide
Habitat	Benthic mat. Irrigation canal from Nile River, Egypt	Terrestrial. Seep walls	Terrestrial	Marine	On rock in a lake	Terrestrial

**Table 2 toxins-16-00451-t002:** Morphological and habitat comparisons between *P. niliensis* sp. nov. and other *Pegethrix* species.

	*P. niliensis* sp. nov.	*P. bostrychoides*	*P. olivacea*	*P. atlantica*	*P. indistincta*	*P. sichuanica*	*P. convoluta*	*P. qiandaoensis*
Trichome/filaments mode of growth	Erect fascicles from the bottom of the culture flasks (liquid medium)	Radial fasciculation, penetrating the agar	Spreading radially, flat and mucilaginous or mounded	Not described	Not described	Radially spreading, with loose fasciculation or clustered	Radially spreading, growing into the agar	Not described
Filaments	Long, wavy, intensely entangled or forming fascicles	Long or short, sometimes forming nodules or loosely to tightly spirally coiled	Long or short, frequently irregularly bent due to uneven cell division along filament. Sometimes loosely coiled to form irregular nodules	Long, fasciculate, straight or loosely coiled	Long, with variation in width between young and mature trichomes. Rarely with more than one trichome sharing a common sheath	Straight or slightly bent	Fasciculate, long, straight or slightly bent, frequently forming loose to compact nodules	Long. Not forming nodules
Trichomes	Not tapered. Apical cells rounded	Not tappered	Cell division along trichomes often irregular, producing cells with variable shape and width	Not tapered	Not tapered. Apical cells rounded	Apical cells rounded	Not tapered	Not tapered. Apical cells rounded
False branching	Not present	Rare, single	Yes	Rare, single	Rare, single or double	Not described	Sometimes singly or doubly false branched	Single or double
Constrictions at cell walls	Facultative	More or less constricted at the distinctly visible cross-walls	Constricted at indistinctly visible cross-walls	Not or slightly constricted at the visible cross-walls	Not or slightly constricted at distinctly visible cross-walls	Not or slightly constricted at the cross-walls	Not or slightly constricted at distinctly visible cross-walls,	Not or slightly constricted at visible cross-walls
Sheaths	Firm, colorless, tightly embracing the trichome (up to 1 μm) or widened (up to 3 μm)	Firm, colorless, usually attached to trichome, occasionally softer, widened, sometimes irregular and stratified	Sheath firm, colorless, usually attached to trichome, occasionally widened	Firm, colorless, attached to the trichome	Firm, usually attached to trichome, occasionally widened, rarely irregular and stratified, absent in immature filaments	Firm, colorless, thin, usually attached to the trichome, occasionally distinct, clear, but occasionally widened	Firm, colorless, usually attached to trichome, occasionally widened, rarely irregular and stratified	Firm, attached to the trichome, occasionally widened
Cell shape	Isodiametric, longer or shorter than wide	Slightly shorter than wide to longer than wide	Cells occasionally isodiametric, shorter than wide in meristematic regions	Mostly shorter than wide	Isodiametric, often shorter than wide especially in meristematic zones, slightly longer than wide in young trichomes	Isodiametric or slightly shorter than wide	Slightly shorter than wide to longer than wide	Isodiametric, longer or shorter than wide
Cell content (light microscopy)	Homogenous or with granule	Rarely with a single central granule	Large central granule	Sometimes with a unique central granule	Not described	Polyphosphate body commonly visible in nucleoid region	Sometimes with a single central granule	Not described
Necridia	Yes	Yes	Yes	Yes	Yes	Yes	Yes	Not described
Cell dimensions (μm)	2.1–3.1 long × 1.9–2.8 wide	1.0–3.0 long × 1.5–2.5–(3.0) wide	1.7–2.6 long × 1.9–3.5 wide	2.4 long × 1.8–3.0 wide	(1.3)–1.7–2.7 long × 1.9–3.3 wide		1.0–2.5–(3.7) long × 1.3–2.5 (3.2) wide	(1.3)1.7–2.7 long × 2.3–4.0 wide
Habitat	Benthic mat. Irrigation canal from Nile River, Egypt	Terrestrial. Sandstone seep wall, UT, USA	Terrestrial. Sandstone seep wall, UT, USA	Rocky substrate over lakes, Azores, Portugal	Terrestrial. Seep wall and waterfall in Navajo Sandstone, UT, USA	Terrestrial. Brick wall alongside mountain, Sichuan, China	Terrestrial. Large seep wall and waterfall in Navajo Sandstone	Planktonic. Freshwater

## Data Availability

The 16S rRNA gene and 16S-23S ITS sequences are available in GenBank (NCBI) under the ID PQ314731.

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
