# Peer review of "Description of Pegethrix niliensis sp. nov., a Novel Cyanobacterium from the Nile River Basin, Egypt: A Polyphasic Analysis and Comparative Study of Related Genera in the Oculatellales Order"

_toxins, 2024, doi:10.3390/toxins16100451_

Round 1
Reviewer 1 Report
Comments and Suggestions for Authors
What makes a species? What makes a genus? The answers are different, and there is no consensus for the latter. The determination of a genus is often subjective. The species within the genus are genetically related, not necessarily similar in functionality. In that sense, my feeling is that we don't have to pursue the perfection in genus determination, and I like the idea to resurrect the old Elainella genus, I wish people had not been too proactive to create new genus based on limited findings in the first place.
The major contribution of the manuscript was to give a clear identification to a new species found in an irrigation canal of the Nile River, Egypt.
Author Response
Reviewer comments: What makes a species? What makes a genus? The answers are different, and there is no consensus for the latter. The determination of a genus is often subjective. The species within the genus are genetically related, not necessarily similar in functionality. In that sense, my feeling is that we don't have to pursue the perfection in genus determination, and I like the idea to resurrect the old Elainella genus, I wish people had not been too proactive to create new genus based on limited findings in the first place.
The major contribution of the manuscript was to give a clear identification to a new species found in an irrigation canal of the Nile River, Egypt.
Authors: The reviewer does not ask for changes. However, we agree with the reviewer and think that this is a very necessary discussion.
Reviewer 2 Report
Comments and Suggestions for Authors
The paper provides valuable data on a toxin-producing strain from a canal nearby the Nile River. Despite that authors themselves realize the need of further cohesion of the discussed genera Pegethrix, Elainella, Maricoleus, Cartusia and Drouetiellaunder the name Elainella, they still describe a new species from the genus Pegethrix. At the moment, according to the current genetic data, this description can be accepted. Some comments are provided below.
lines 30-32: "Despite their widespread distribution and substantial eco-30 logical contributions, cyanobacteria have a complicated taxonomy, with many cryptic taxa having been misclassified [6]" - Although there is a reference, i would strongly recommend to edit this sentence because the broad widespread in nature has nothing to do with complicated taxonomy.
Line 75: it is recommended to add the dimensions of the sheaths, and to point are they tightly embracing the trichome, or are widened (as it is described further on in Table 2)
Line 76: I would suggest to replace " trichomes ...with rounded ends" with "apical cells rounded, without calyptra". The same for line 137 and for texts in Table 1
Table 2: I would recommend to replace "thallus" in this table with "trichome/filaments mode of growth" or something similar, since thallus is a term for a vegettaive algal body and includes one sheath with one trichome in case of this species - P. niliensis
Line 270:"In conclusion, our paper represents a significant contribution to biodiversity research" - authors easily can take this sentence away and led the readers to say this :). The same for lines 276-278 - "valuable insights" should be taken away
Author Response
Reviewer comments: The paper provides valuable data on a toxin-producing strain from a canal nearby the Nile River. Despite that authors themselves realize the need of further cohesion of the discussed genera Pegethrix, Elainella, Maricoleus, Cartusia and Drouetiellaunder the name Elainella, they still describe a new species from the genus Pegethrix. At the moment, according to the current genetic data, this description can be accepted. Some comments are provided below.
lines 30-32: "Despite their widespread distribution and substantial eco-30 logical contributions, cyanobacteria have a complicated taxonomy, with many cryptic taxa having been misclassified [6]" - Although there is a reference, i would strongly recommend to edit this sentence because the broad widespread in nature has nothing to do with complicated taxonomy.
Authors: we agree.
Line 75: it is recommended to add the dimensions of the sheaths, and to point are they tightly embracing the trichome, or are widened (as it is described further on in Table 2)
Authors: we agree
Line 76: I would suggest to replace " trichomes ...with rounded ends" with "apical cells rounded, without calyptra". The same for line 137 and for texts in Table 1
Authors: we agree
Table 2: I would recommend to replace "thallus" in this table with "trichome/filaments mode of growth" or something similar, since thallus is a term for a vegettaive algal body and includes one sheath with one trichome in case of this species - P. niliensis
Authors: we agree
Line 270:"In conclusion, our paper represents a significant contribution to biodiversity research" - authors easily can take this sentence away and led the readers to say this :). The same for lines 276-278 - "valuable insights" should be taken away
Authors: we agree